# AI-based detection and classification of anomalous aortic origin of coronary arteries using coronary CT angiography images

Isaac Shiri[1], Giovanni Baj [1,5], Pooya Mohammadi Kazaj [1,5], Marius R. Bigler[1], Anselm W. Stark[1], Waldo Valenzuela [2], Ryota Kakizaki[1], Matthias Siepe[3], Stephan Windecker[1], Lorenz Räber[1], Andreas A. Giannopoulos[4], George CM. Siontis[1], Ronny R. Buechel [4] & Christoph Gräni [1] ✉

Anomalous aortic origin of the coronary artery (AAOCA) is a rare cardiac condition that can lead to ischemia or sudden cardiac death, yet it is often overlooked or falsely classified in routine coronary CT angiography (CCTA). Here, we developed, validated, externally tested, and clinically evaluated a fully automated artificial intelligence (AI)-based tool for detecting and classifying AAOCA in 3D-CCTA images. The discriminatory performance of the different models achieved an AUC ≥ 0.99, with sensitivity and specificity ranging 0.95-0.99 across all internal and external testing datasets. Here, we present an AI-based model that enables fully automated and accurate detection and classification of AAOCA, with the potential for seamless integration into clinical workflows. The tool can deliver real-time alerts for potentially high-risk AAOCA anatomies, while also enabling the analysis of large 3D-CCTA cohorts. This will support a deeper understanding of the risks associated with this rare condition and contribute to improving its future management.

Anomalous Aortic Origin of the Coronary Artery (AAOCA) is a rare congenital heart disease that presents in various forms[1]. Despite the low prevalence, it affects approximately 1.3 million people in the United States[2,3]. While most AAOCA, especially those with a pre-pulmonic or retro-aortic course of the anomalous vessel, are considered low-risk anatomy[4,5], those with an intramural and interarterial course (i.e., between the great arteries of the aorta and the pulmonary artery) are referred to as high risk as they potentially lead to ischemia and adverse cardiac events[5]. In fact, AAOCA has been associated with up to 20% and 30% of sudden/ unexpected deaths among young athletes and military recruits who engaged in intense physical activity, as revealed in autopsy reports, respectively[3,6–9]. Therefore, the correct detection is crucial for optimal management of the disease[5,10,11].

AAOCA can be also incidentally discovered during anatomical imaging studies performed for other conditions, such as the diagnosis of coronary artery disease (CAD)[12]. As coronary computed tomography angiography (CCTA) is recommended as the first-line noninvasive anatomical imaging modality, according to guidelines from the US, and Europe for the detection of possible CAD, the absolute number of detected AAOCA is increasing[4,10,11,13]. In clinical routine settings, AAOCA may still be overlooked during CCTA scans, especially as smaller centers perform more of these tests and may not be as experienced in detecting this rare disease. Moreover, due to unfamiliarity, imagers often struggle with accurately classifying anomalies as high-risk or low-risk (historically referred to as 'malignant' and 'benign'). Therefore, there is an unmet need for automated tools to analyze CCTA images to reduce the risk of overlooking rare possible high-risk AAOCA. Such

[1]Department of Cardiology, Inselspital, Bern University Hospital, University of Bern, Bern, Switzerland. [2]University Institute for Diagnostic and Interventional Neuroradiology, Inselspital, Bern University Hospital, University of Bern, Freiburgstrasse, Bern, Switzerland. [3]Center for Congenital Heart Disease, Department of Cardiac Surgery, Inselspital, Bern University Hospital, University of Bern, Bern, Switzerland. [4]Department of Nuclear Medicine, Cardiac Imaging, University Hospital Zurich, Zurich, Switzerland. [5]These authors contributed equally: Giovanni Baj, Pooya Mohammadi Kazaj. ✉e-mail: christoph.graeni@insel.ch

tools could also help to analyze large available databases to increase the detection of AAOCA in retrospective or prospective datasets and to link the anatomical variants with outcomes to better understand the lifetime risk. Artificial intelligence (AI)-based tools have been developed for 3D-CCTA image analysis, including automated coronary centerline detection, coronary segmentation, automated classification and measurement of plaque, and automated reporting, but none of these tools are available for automated AAOCA detection[14,15].

In this work, we developed and evaluated a fully automated AI-based screening tool for detecting and classifying AAOCA in 3D-CCTA images. The tool employs a two-step deep learning algorithm for segmentation/cropping and classification. Following image cropping, the classification step focuses on detecting AAOCA cases, determining whether the anomalous vessel is right (R-AAOCA) or left (L-AAOCA), and categorizing the courses as either high or low anatomical risk. This tool can operate seamlessly alongside clinical assessments of CCTA scans, providing real-time alerts to medical personnel about potential high-risk AAOCA anatomy, which is important in a rare disease where physicians are unfrequently exposed. Furthermore, it can be utilized to identify AAOCA in large CCTA patient cohorts, given that the risk associated with the disease is currently unknown and warrants future investigation. Therefore, this tool could potentially enhance diagnostic efficiency, assist in managing AAOCA, and improve outcomes in these patients.

## Results

### Dataset
Overall, we excluded from the analysis 14 patients without contrast-enhanced cardiac CT images, 10 with uninterpretable images due to severe artifacts, 6 patients with images with cropped regions of the aorta, and 12 without a report for the origin and anatomical risk classification. Figure 1 shows descriptive information of the individuals and images in each dataset, and Supplementary Table 1 provides details of the different datasets used for model training and evaluation. Supplementary Tables 2 and 3 provide detailed demographic information, scanner specifications, and image acquisition parameters across the different datasets. For the AAOCA detection, 2376 patients (4128 CCTA images) were included, with 335 AAOCA patients (1056 CCTA images). Out of the entire dataset, 998 patients (998 CCTA images) did not have labeled cases for AAOCA, which corresponds to the external clinical evaluation dataset. For the anomalous coronary artery classification task, 327 patients (1028 CCTA images) were included, with 132 L-AAOCA patients (322 CCTA images). For the risk classification tasks, 326 patients (1025 CCTA images) were included, with 225 high-risk anatomy patients (820 CCTA images). Out of 225 CCTA high-risk anatomy AAOCA, 208 patients (773 CCTA images) had an inter-arterial course, and 17 patients (47 CCTA images) had a sub-pulmonic course.

### Segmentation
The segmentation model for the aorta and LV achieved a mean dice score of 0.89 (0.83 for myocardium, 0.90 for LV cavity, and 0.93 for aorta) in 5-fold cross-validation. Using this segmentation model, we cropped all images to the desired size. Although we needed only a rough segmentation for cropping purposes, we checked all segmented and cropped images across different datasets and found no miscropping. Examples of segmentation were provided in Supplementary Fig. 1.

### Classification
Summary results of different detection and classification tasks for two different testing sets (internal and external testing datasets) for the ensemble model of 5-fold are presented in Table 1 (sex-specific analyses are detailed in Supplementary Tables 4 and 5). The ROC-AUC of the different models was higher than 0.99 across all testing datasets and models. For the detection of AAOCA, a sensitivity of 0.99 and 0.96 was achieved for the internal and external testing datasets,

respectively. The specificity for AAOCA detection and origin classification was higher than 0.99 across all testing datasets. Sensitivity and specificity in anatomical risk classification were both higher than 0.96 in both testing datasets. The detailed results of different metrics for different models across various test datasets, including the mean of 5-folds, the metrics for each fold, and sex are provided in Supplementary Tables 4–14.

The ROC curves for the five different folds and the ensemble model for different detection and classification tasks across different testing datasets (internal and external testing datasets) are provided in Fig. 2 (sex-specific ROC curves are provided in Supplementary Figs. 2 and 3). Supplementary Figs. 4–6 show the ROC curve for the mean of the five folds. Figure 3 presents the confusion matrix for the classification models in different detection and classification tasks across different testing datasets for the ensemble model, using a cut-off of 0.5 based on the training and internal validation dataset (sex-specific confusion matrix are provided in Supplementary Figs. 7 and 8). Supplementary Figs. 9–17 illustrate the confusion matrices for classification in different detection and classification tasks and sex with cut-offs ranging from 0.1 to 0.9. These figures demonstrate that by decreasing the cut-off, the number of true and false positives could increase to some extent, allowing for a higher sensitivity and lower specificity model depending on clinical use. Different metrics for different cut-offs were provided in Supplementary Tables 15–23 for different detection and classification tasks and sex.

### Interpretability
Figure 4 illustrates different cases, including normal and AAOCA, with various anomalies and the corresponding GradCam + + feature overlays on the CCTA images. As shown in this figure, the networks learned the exact pattern of the coronary arteries, and their activation maps indicate the location and course of different coronary arteries.

### Feature space visualization using t-SNE
Figure 5 shows the 2D t-SNE maps of the latent features extracted from the last layer of one of the models trained for the anomaly detection task, with each point representing an image from different datasets (sex-specific t-SNE maps are provided in Supplementary Figs. 18 and 19). This figure was generated based solely on the latent feature set of an image, without including any additional information such as labels, centers, or datasets in the plotting process. Labels and dataset types were used only for plotting and coloring purposes. As shown in Fig. 5a, there is no center or data-based clustering, indicating the model's generalizability without bias toward any specific center or database. Moreover, when colored based on normal and AAOCA cases, it shows two distinct clusters that differentiate between the two classes, with only a few cases misclassified. Figure 5b, c displays the different coronary arteries and the risk classification of their course in AAOCA cases based on anomaly detection. This figure demonstrates that by only coloring the different classes, we can observe distinct clusters, indicating that the network has learned the pattern of AAOCA.

### Screening
Figure 6 shows different cases from the external clinical evaluation dataset that were detected as AAOCA and confirmed by a physician. As shown in this figure, one high take-off (R-AAOCA, low-risk anatomy), two R-AAOCA (high-risk anatomy), and one L-AAOCA (left circumflex, low-risk anatomy) were detected and confirmed out of 24 cases flagged as anomalies (4 true positives) out of 998 patients. In all four true positive cases, origins and risks were correctly classified by the models. Moreover, we highlighted challenging cases that were correctly identified as normal, including those highly affected by motion artifacts. Despite their high similarity to the different AAOCAs in different slices, the network could classify them correctly. We also present false

positive cases, which were detected as AAOCA by the networks but were not confirmed as AAOCA by the physician.

### Real-world use case

Supplementary Figs. 20–25 show the results from different strategies, indicating a slight improvement in performance with no significant differences when using larger data for training. When using more datasets for training, we did not find more positive cases from the external clinical evaluation dataset; however, the number of false positives decreased from 20 to 10 and 9 in the external clinical evaluation dataset in Strategy 2 and 3. The model with more training data

sets was more likely to produce generalizable results in real-world scenarios across other centers, as it utilizes all labeled datasets.

## Discussion

In this study, the fully automated tool for detecting and classifying AAOCA, utilizing a two-step deep learning algorithm for segmentation, cropping, and classification, demonstrated its ability to accurately identify anomalies, determine the affected coronary artery, and classify courses as high or low risk. It performed robustly on internal and external test sets and successfully flagged unlabeled anomalies in a real-world dataset, later confirmed by cardiologists, highlighting its

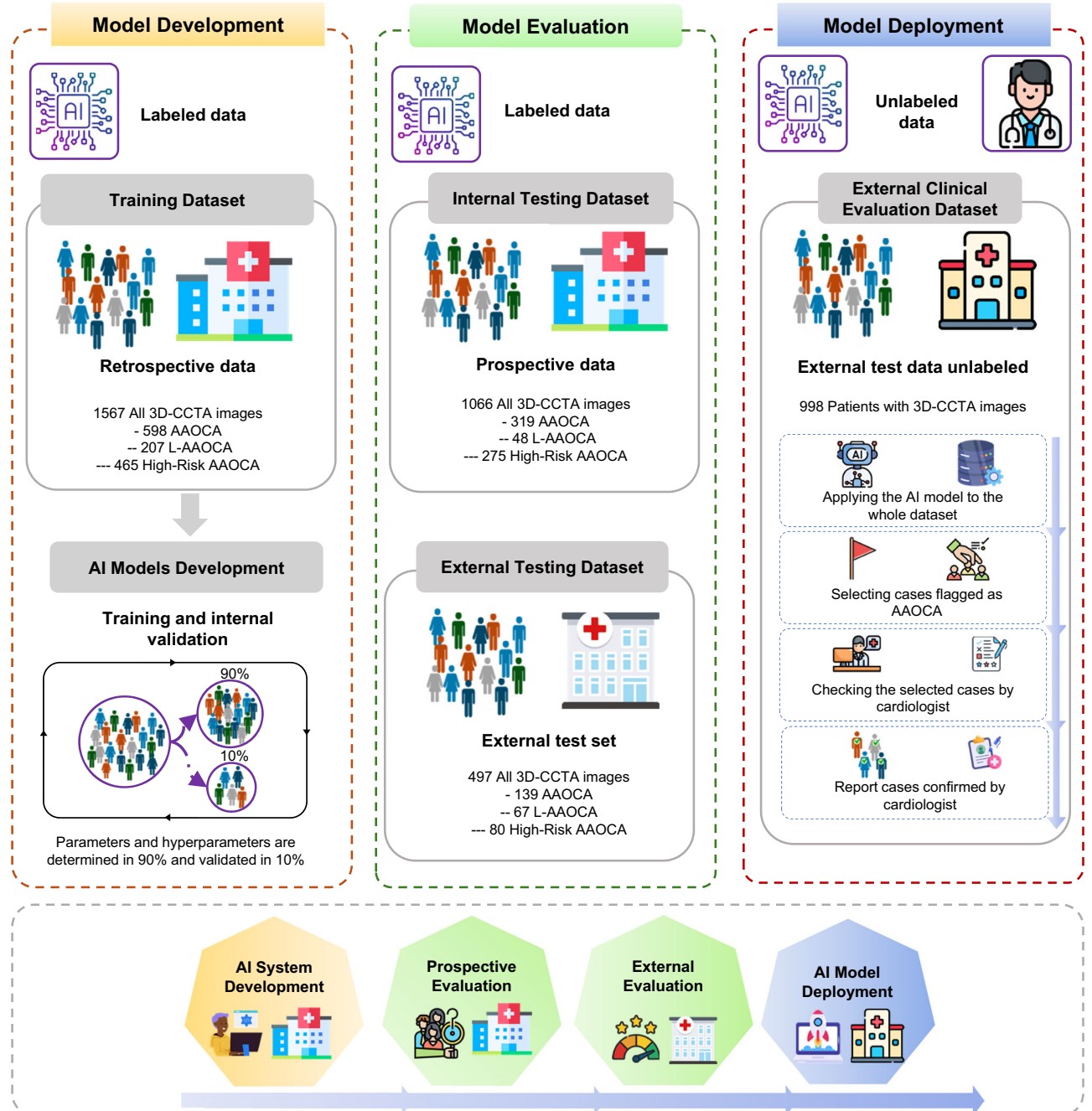

**Fig. 1 | Summary of model development and evaluation throughout the entire study.** The model was developed using a retrospective dataset from the training and internal validation dataset and tested on a prospective cohort of the internal testing dataset from the same center. Models were then externally validated (external testing dataset). We finally simulated a real-world scenario using an unlabeled dataset (external clinical evaluation dataset). AAOCA Anomalous Aortic Origin of the Coronary Artery, CCTA coronary CT angiography, L-AAOCA left AAOCA, R-AAOCA right AAOCA.

**Table 1 | Summary of different classification metrics for the ensemble models for different test datasets**

| Metrics | Anomaly detection | | Origin classification | | Risk classification | |
|---|---|---|---|---|---|---|
| | **Testing dataset** | | **Testing dataset** | | **Testing dataset** | |
| | Internal | External | Internal | External | Internal | External |
| ROC AUC | 0.998 | 0.999 | 0.999 | 0.999 | 0.999 | 0.996 |
| Sensitivity | 0.987 | 0.957 | 0.938 | 0.955 | 0.989 | 0.962 |
| Specificity | 0.989 | 1.000 | 1.000 | 1.000 | 1.000 | 0.963 |
| F1-score | 0.981 | 0.978 | 0.968 | 0.977 | 0.995 | 0.969 |
| PPV | 0.975 | 1.000 | 1.000 | 1.000 | 1.000 | 0.975 |
| AUPR | 0.996 | 0.999 | 0.997 | 0.999 | 1.000 | 0.997 |
| Accuracy | 0.989 | 0.988 | 0.990 | 0.978 | 0.990 | 0.963 |

Anomaly detection: distinguishing between normal cases and those with AAOCA; Origin classification: classifying the anomalous vessel into either the right (R-AAOCA) or left (L-AAOCA; Risk classification: scoring the AAOCA risk, classifying it as either low-risk or high-risk anatomy. AAOCA: Anomalous aortic origin of the coronary artery, ROC: Receiver Operating Characteristic, AUC: Area under the curve, F1-score: a measure of a test's accuracy, the harmonic means of precision and recall, PPV: Positive predictive value, AUPR: Area under the precision-recall curve. The slight differences in ROC-AUC values between figures and tables are due to the use of different libraries for calculations and rounding discrepancies. All values are rounded to three decimal places, which may result in a value of 1. Source data are provided as a Source Data file.

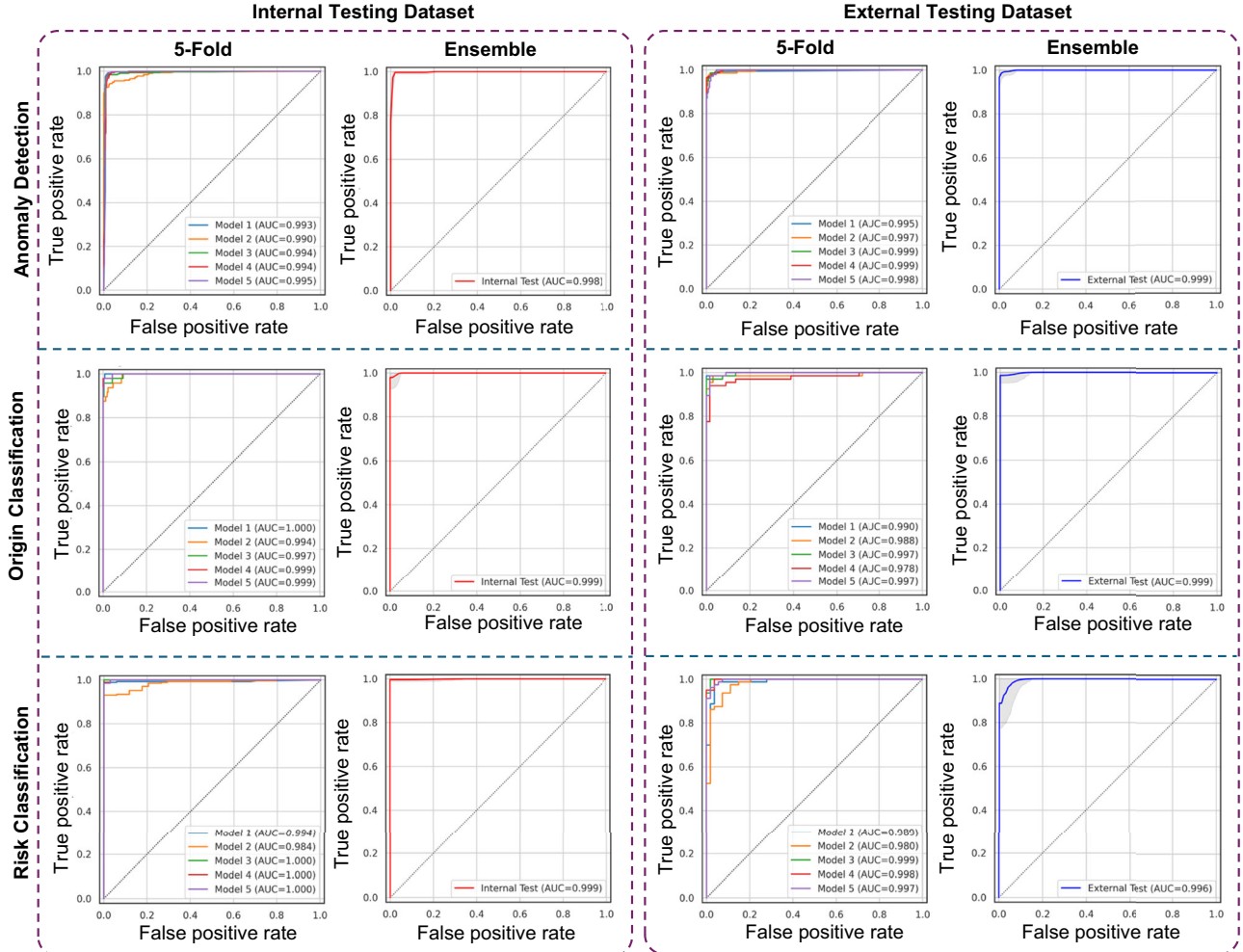

**Fig. 2 | ROC curves including the 5 different folds and the ensemble of 5 models across different tasks for various test datasets.** Confidence intervals and tolerance intervals for the ensemble models were computed with the bootstrap method (10,000 iterations), the gray area on the ensemble figures is the tolerance interval. Anomaly Detection: distinguishing between normal cases and those with AAOCA; Origin Classification: classifying the anomalous vessel into either the right (R-AAOCA) or left (L-AAOCA); Risk Classification: scoring the AAOCA risk, classifying it as either low-risk or high-risk anatomy. AUC: area under the curve. Source data are provided as a Source Data file.

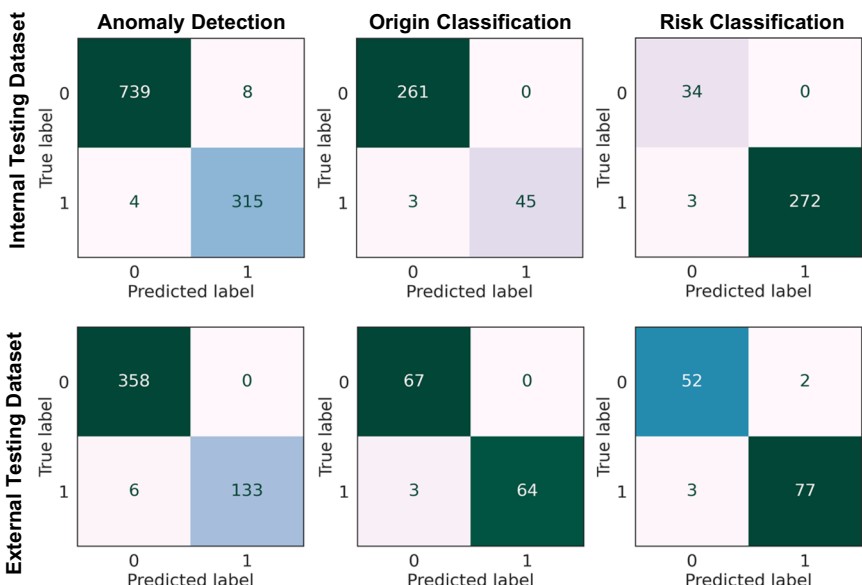

**Fig. 3 | Confusion matrices of different models in various Tasks for different datasets in the ensemble model.** Anomaly Detection: distinguishing between normal cases and those with AAOCA; Origin Classification: classifying the anomalous vessel into either the right (R-AAOCA) or left (L-AAOCA); Risk Classification: scoring the AAOCA risk, classifying it as either low-risk or high-risk anatomy. Source data are provided as a Source Data file.

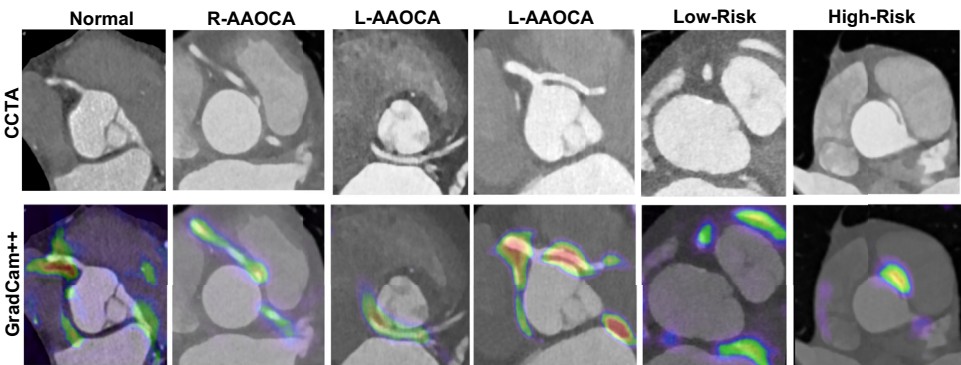

**Fig. 4 | CCTA images and corresponding GradCam + + features for normal cases and different anomalies at block 2 (out of 4) in the Anomaly Detection model.** AAOCA Anomalous Aortic Origin of the Coronary Artery, CCTA Coronary CT Angiography, R-AAOCA Right AAOCA, L-AAOCA left AAOCA, Low-risk: pre-pulmonic course, High-risk: inter-arterial course. GradCam: Gradient-weighted class activation mapping.

effectiveness in clinical practice. This tool can assist in the real-time detection of AAOCA during CCTA analysis, reducing human error and providing valuable support in identifying cases that might otherwise be overlooked during manual evaluation. Furthermore, this automated tool will facilitate the assessment of large retrospective and prospective datasets, while linking anatomy to outcomes will enhance risk stratification and inform the future management of AAOCA.

Accurate detection of AAOCA in CCTA is critical for optimal management, as those with high-risk anatomical features are potentially leading to hemodynamically relevance[6–9]. This is particularly important as CCTA is now established as a first-line non-invasive imaging technique for diagnosing suspected CAD. This will lead to increased detection of absolute numbers of AAOCA cases, highlighting the need to differentiate between low-risk coincidental bystanders and high-risk cases that require further functional downstream testing. However, AAOCA may still be overlooked in routine clinical settings, especially as imagers may be rarely confronted with this entity. This is emphasized by the fact that, although AAOCA is generally considered to have a very low prevalence, it may be more common than previously thought in the current area[4,10,11,13,16–19]. In addition, the difficulty in correctly classifying anomalies as potentially high risks versus low risks needs expertize, which may be lacking in smaller centers providing CCTA programs. Consequently, AI tools could potentially help to alert physicians about AAOCA and its potential risks.

Furthermore, AAOCA has been associated with adverse events and sudden cardiac death, particularly during physical activity, as evidenced by autopsy reports[6–9]. However, autopsy reports are associated with a selection bias towards potentially high-risk patients and do not represent the risk for living individuals with AAOCA[20]. Therefore, there is an unmet need to analyze large retrospective and prospective CCTA datasets, such as CONFIRM 2[21], to assess the true prevalence and risk of AAOCA when linked to outcomes. Our AI tool facilitates the analysis of large datasets, enhancing AAOCA detection and improving risk stratification by correlating anatomical findings with clinical outcomes.

Although AI-based tools have been developed for various aspects of CCTA image analysis, none is currently available for automated AAOCA detection, especially due to the lack of proper ground truth[14,15]. A recent study[22] proposed an automated deep learning-based segmentation and detection method for AAOCA, focusing on the

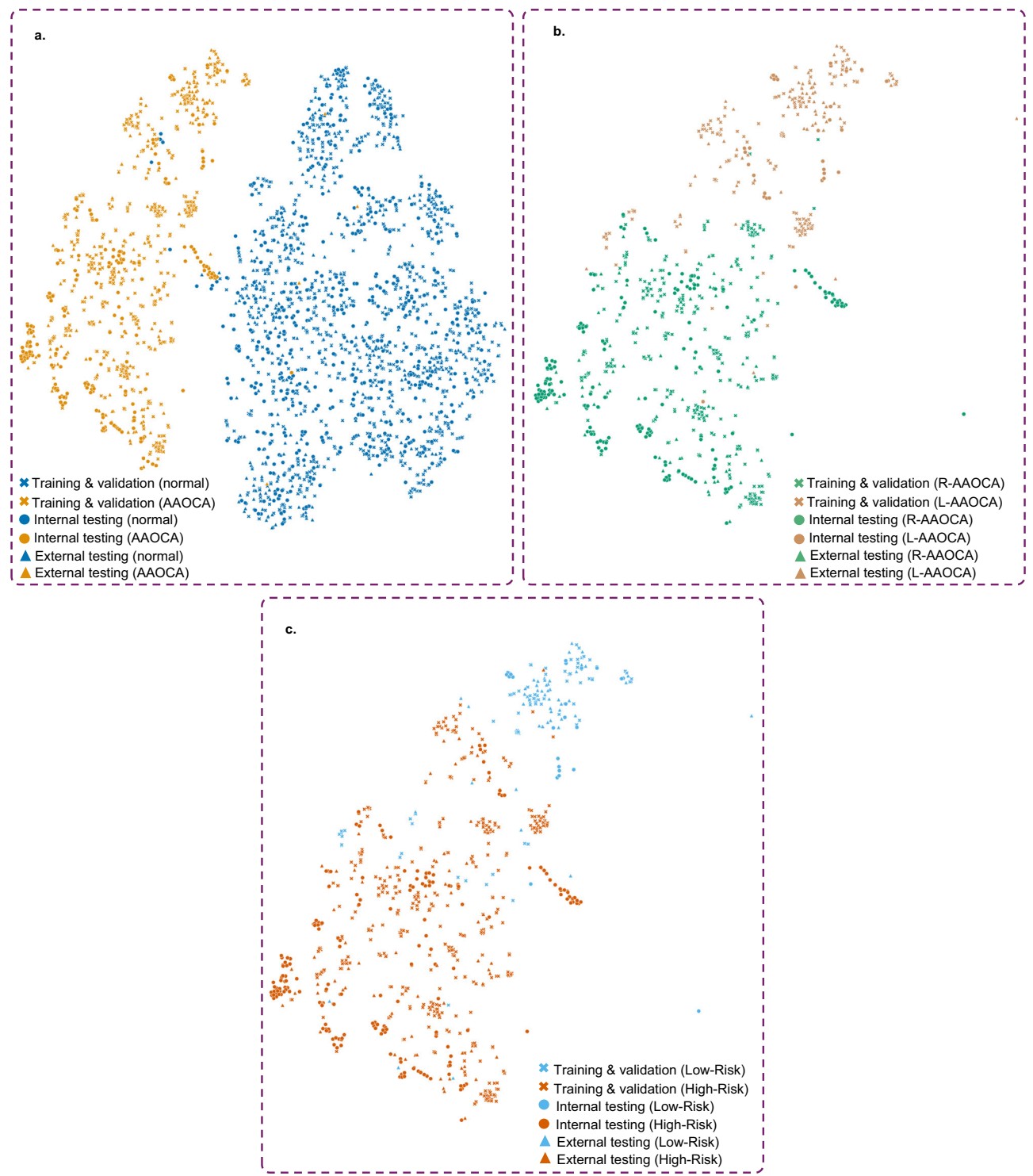

**Fig. 5 | The 2D t-SNE maps of latent features extracted from the last layer of the anomaly detection model. a** The t-SNE map of the anomaly detection model colorized for anomalies and normal cases, **b** the t-SNE map of the anomaly detection model colorized for right and left anomalies (only for the anomaly dataset), and **c** the t-SNE map of the anomaly detection model colorized for high and low anatomical risk anomalies (only for the anomaly dataset). The cross shape represents the training and validation datasets, the circle shape represents the internal testing dataset, and the triangle shape represents the external testing dataset. t-SNE t-distributed stochastic neighbor embedding, AAOCA anomalous aortic origin of the coronary artery, CCTA coronary CT angiography, R-AAOCA right AAOCA, LAAOCA left AAOCA. Source data are provided as a Source data file.

segmentation of the aorta and coronary arteries. They used a small, single-center dataset to develop a segmentation model based on 124 CCTA scans, reporting an accuracy, precision, and recall of 1 in a test set comprising only 13 images. Their method relies on the segmentation of coronary arteries using multi-view 2D segmentation of CCTA,

followed by post-processing of the segmentation output, rule-based post-processing such as connectivity, and characteristic analysis of anomalous using a decision tree model. Although the authors implemented interesting approaches, the study has several limitations. Firstly, the small single-center dataset set for training, evaluation, and

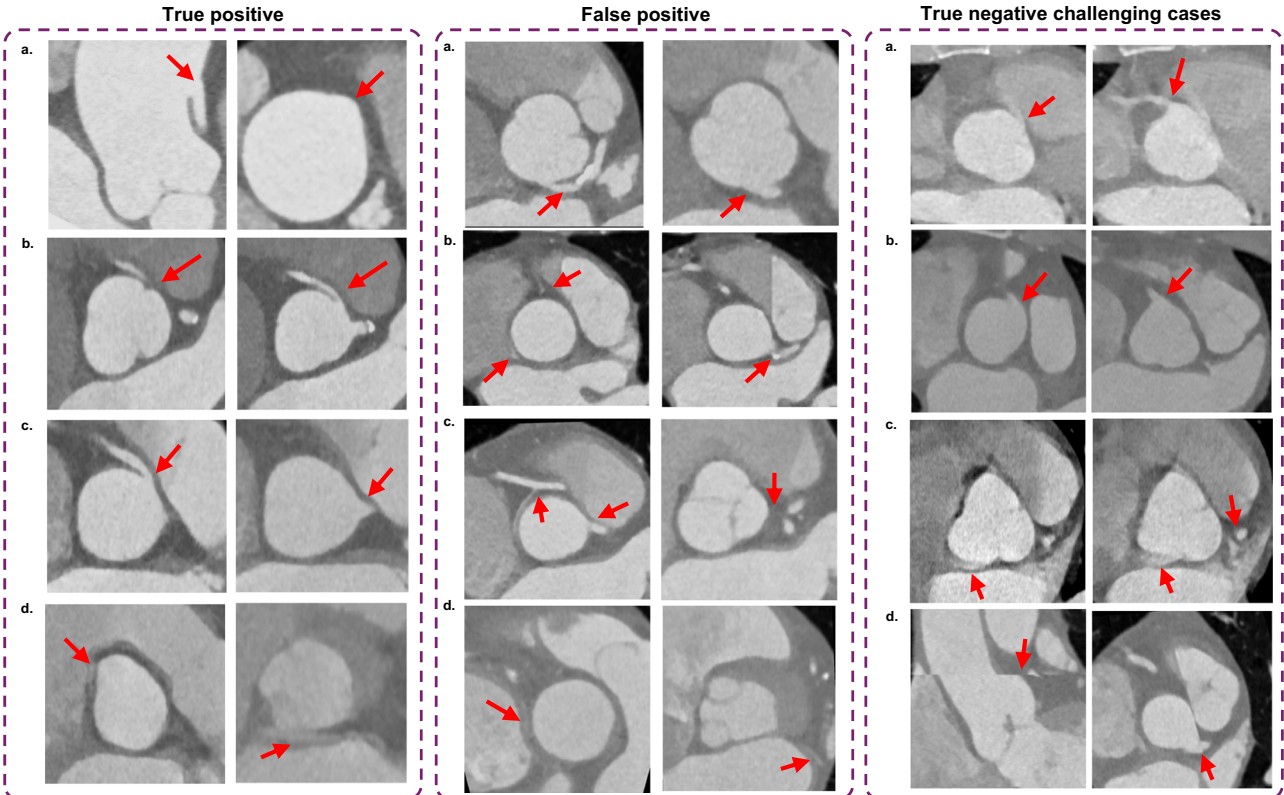

**Fig. 6 | Developed model applied on external clinical evaluation dataset (unlabeled dataset) for real-world screening scenario.** True positives: **a**) High take-off of the right coronary artery (R-AAOCA) with low-risk anatomy. **b**, **c**) Right coronary artery originating from the left coronary sinus (R-AAOCA) with high-risk anatomy. **d** Circumflex arteries originating from the right coronary sinus (L-AAOCA, left circumflex) with low-risk anatomy. False positives: **a**) Left coronary artery originating within the left coronary sinus but very close to the non-coronary sinus (**b**) Appearance of a very thin coronary artery (possible conus artery) mimicking an R-AAOCA and possible contrast agent artifact mimicking L-AAOCA (left circumflex), **c**) Very thin and short possible conus artery from the right coronary artery mimicking a left circumflex anomaly with immediate disappearance, **d**) Resembling of a very faint coronary artery (possibly conus artery) resembling the left circumflex (L-AAOCA), however, there was a large left circumflex from the main stem present. True negative challenging cases: **a**, **b**) Motion artifact creating an R-AAOCA-like anomaly, **c**) Motion artifact generating an L-AAOCA-like (left circumflex) anomaly, **d**) Motion (step artifact) artifact removing the connection of the left coronary artery to the left sinus.

testing, without any external test set, limits the model's robustness and generalizability. Moreover, the segmentation model was trained on only 99 images, with limited variability across anomalies. Anomalies can vary significantly within datasets, and the segmentation model could easily fail on other anomalies as even the normal coronary arty segmentation is a challenging task despite the availability of a large dataset[23]. As the detection of anomalies relies on the segmentation model, any mis-segmentation could lead to failures in the classification model. In contrast, our study addresses these limitations by employing a rough segmentation of larger structures, used solely for cropping purposes. We utilized a larger training dataset to train a classification model based on a deep learning network. We internally and externally tested the model with a large dataset, demonstrating its effectiveness in real clinical situations. In addition, we developed two more models that classify the anomalous coronary artery and its risks. Our pipeline can be used in fully automated approaches as well as in physician-in-the-loop approaches.

While this AI tool has the potential to assist by alerting physicians in real-time to the presence of AAOCA and its potential risks, in smaller centers with less experienced readers, reliance on AI without expert oversight might lead to false positives, causing unnecessary follow-ups, increased patient anxiety, and potential harm. In high-volume centers with experienced readers, the tool could serve as an effective safety check to ensure no cases of anomalous coronaries are missed. Furthermore, the current model is designed to augment—not replace—clinical expertize, particularly in settings where additional support can

aid in efficiently identifying AAOCA cases. Given the rarity of AAOCA, minimizing missed cases is crucial, emphasizing the importance of accurate true positive detection. While false positives were observed in the external clinical test set, they were significantly reduced from 2% to 0.9% through more training datasets without compromising true positive detections (from Strategies 1 to 3). As our study demonstrates, further training on larger datasets could reduce false positives even further. Future research should focus on enhancing these models to minimize false positives and facilitate their integration into clinical practice

The current study bears some limitations due to the low prevalence of specific anomalies; for instance, conditions like anomalous left coronary artery from the pulmonary artery are not included in our training dataset. We hypothesize that the network still recognizes such cases as anomalies based on learned patterns of normal coronary arteries. This shows the potential for future studies to use unsupervised learning or out-of-distribution analysis to better manage anomaly types not represented in the training data. The model development and evaluation were performed using CCTA images. However, with further testing, this model could potentially be adapted for non-coronary contrast-enhanced CT images in future studies, broadening its applicability to other imaging acquisition and protocols toward enhancing AAOCA detection. Developing a deep learning model that detects anomalies in non-contrast images, such as standard chest CTs, would expand its utility beyond cardiac imaging to include applications like any chest CT scans. However, detecting coronary

arteries in non-contrast images presents significant challenges due to low image contrast and possibly low resolution, complicating the analysis even for highly experienced cardiologists and radiologists. While our model has undergone rigorous testing, including internal prospective tests, external testing datasets, and external clinical evaluation datasets to simulate real-world applications, it has yet to be implemented in a real clinical setting. To fully evaluate its practical utility, this AI tool should be prospectively deployed in real-world clinical settings or assessed in dedicated randomized clinical trials, comparing it to standard-of-care approaches to determine its clinical impact and influence on downstream testing.

## Methods

The design and reporting of this study follows different dedicated guidelines for AI applications in medical imaging (see the Supplementary Method for more details). Figure 7 shows an overview of the entire study implementation, including the datasets, examples of CCTA images, the different networks, and a summary of the results.

### Datasets

This study utilized multiple datasets for different tasks. Initially, to develop the segmentation model, we employed 1203 open-access CT images that included ground-truth segmentations of the aorta and left ventricle (LV)[24]. In addition, data from three different centers were used to develop and evaluate the detection and classification of AAOCA. Bern University Hospital provided both retrospective cohort (training dataset) and prospective cohort (internal testing dataset) datasets. Zurich University Hospital contributed with a single dataset used as an external testing dataset (retrospective and prospective cohort). From these two centers, both normal and AAOCA cases with labels for each type of anomaly, specifying which coronary was anomalous and coding its anatomical course, were included (labeled by two experienced cardiologists specialized in CCTA). The data extraction, anonymization, and classification processes were carried out using the medical-blocks system[25], ensuring standardized and compliant handling of the medical information. The third dataset was an open-access 3D-CCTA dataset (Guangdong Provincial People's Hospital acquired from April 2012 to December 2018)[23], which lacks labeling and was used for the screening functionality of the model in real-world scenarios (external clinical evaluation dataset). Detailed information for each dataset is provided in Supplementary Table 1–3. Information regarding the open-access dataset and the segmentation dataset are available online[23,24]. From all datasets, we excluded patients without contrast-enhanced cardiac CT images, those with uninterpretable images due to severe artifacts, and images with cropped regions of the aorta. From the datasets used for training and testing, we excluded from the analysis those without a report for the origin and anatomical risk classification. All procedures in studies involving human participants adhered to the ethical standards of the institutional and/or national research committee, the 1964 Helsinki Declaration, and its subsequent amendments or comparable ethical standards. The Bern (KEK 2020-00841 and KEK 2021-0058) and Zurich (KEK 2015-0235 and KEK 2014-0632) cantonal ethics committee approved the study design for the dataset used in the current study. Participants in the study provided written informed consent prior to any data collection, and all imaging data were anonymized.

### Segmentation Model

In AAOCA, the origin and proximal course of the anomaly are critical. Therefore, we focused our analysis exclusively on this region compared to the entire 3D-CCTA cardiac image. We developed a fully automated segmentation model to automate cropping in this area. Given the high variability in the anatomy of the aorta and to ensure robust cropping that does not fail in different datasets, we segmented both the aorta and the LV. We employed the nnU-Net[26] network with deep supervision and extensive data augmentation to develop this segmentation model. The models were trained using a 5-fold cross-validation approach, and the ensemble of 5 models was used for inference on the test sets. This ensemble model was then applied across the rest of the dataset for which we had no labeled segmentation (dataset for classification from training, validation, internal and external testing, and external clinical evaluation dataset).

Following the segmentation of the aorta and LV, the central point where the aorta and LV segments intersected was then located. This point was subsequently adjusted by moving it 1 cm to the right and upward. This adjustment was made to ensure that more of the aorta's curved structure would be included in the cropped images. A box with dimensions of $8 \times 8 \times 6 \, \text{cm}^3$ fitted over the image, and the region of interest was cropped.

### Classification Model

**Image preprocessing.** Before being input into the classification models, the cropped images underwent different preprocessing steps. To ensure consistency and prevent data leakage, these preprocessing steps were first defined and implemented using only the training and (internal)-validation datasets. The same procedures were then applied to the testing dataset. Initially, cropped images were resampled using spline interpolation to uniform dimensions of $215 \times 215 \times 85$, determined by the median values in each direction from the training and (internal)-validation dataset. To minimize noise and remove outlier intensities, the resampled images were clipped to a Hounsfield Units range of $-1024$ to $1024$. Lastly, to reduce the dynamic range further, the image intensities were discretized into 256 levels (ranging from 0 to 256) and subsequently normalized to a scale from 0 to 1 with a min-max scaling approach.

**Data augmentation.** In cases where different phases of cardiac images were available, we utilized all available reconstructed phases as a form of natural data augmentation to increase both the dataset size and the model's generalizability. We implemented various data augmentation techniques, including random noise, blurring, and contrast, to mimic different presentations in 3D-CCTA images from different scanners and centers. Moreover, to increase the robustness of our model under conditions of motion and step-and-shoot acquisition in older CCTA scanner generation, which can significantly change image appearance, we simulated these conditions and included them in our augmentations. Examples of different image augmentations are provided in Supplementary Fig. 26.

**Deep-learning modeling.** A 3D Squeeze-and-Excitation residual network (4 blocks) consisting of 154 layers[27] for various detection and classification tasks was implemented. To train the networks, we split the training dataset (patient-wise) into training and (internal)-validation sets with a ratio of 90:10 (Fig. 1). In the data-splitting process, we ensured that the images of the same patient were present only in either training or (internal)-validation. The learning rate was initially set at 0.001 and adjusted based on a cosine annealing scheduler[28], ensuring steady convergence. The network was trained over 300 epochs, adopting binary cross entropy as the loss function with early stopping. The best model was selected based on its performance in reducing training and (internal)-validation loss and its overall accuracy on (internal)-validation datasets. We used different metrics such as accuracy, sensitivity, and specificity, evaluating the models comprehensively during the (internal)-validation phase, ensuring that the chosen model minimized loss and maximized predictive reliability.

**Different classification tasks.** We addressed three distinct classification tasks to enhance the detection and classification of AAOCA, including:

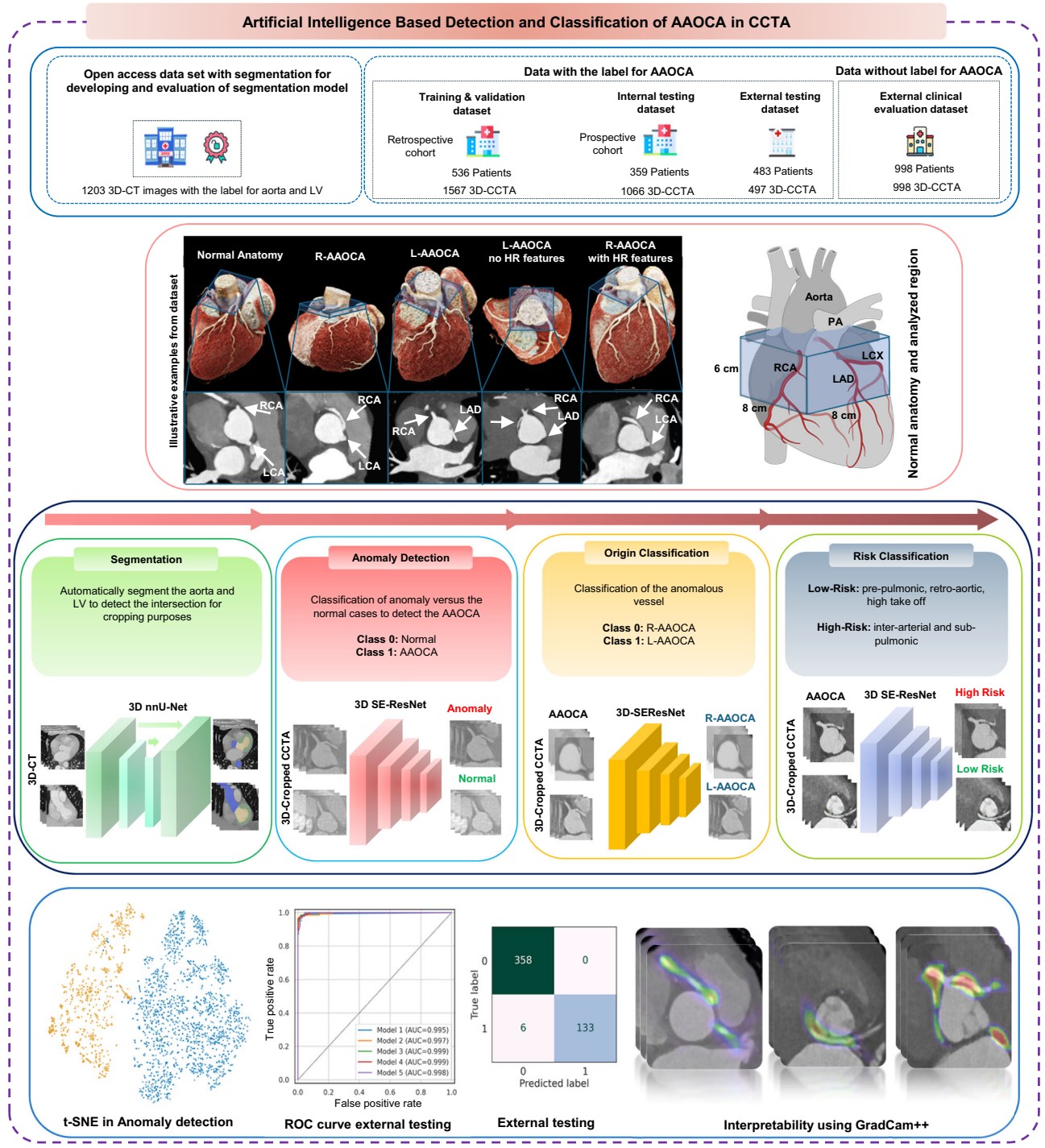

**Fig. 7 | The flowchart of the current study shows an overview of the entire study implementation, including the datasets, examples of CTTA images, the different networks, and a summary of the results.** AAOCA Anomalous Aortic Origin of the Coronary Artery, CCTA coronary CT angiography, RCA Right coronary artery, LAD left anterior descending, LCX left circumflex, PA Pulmonary Artery, HR High Risk, nnU-Net No new U-Net, SE-resNet Squeeze-and-Excitation residual network, t-SNE t-distributed stochastic neighbor embedding, ROC receiver operating characteristic. The schematic of the heart was created in BioRender. Created in BioRender. Shirilord, I. (2025) https://BioRender.com/d49d159.

Anomaly detection: This includes distinguishing between normal cases and those with AAOCA.

Origin classification: This involved classifying the anomalous vessel into either the right (R-AAOCA) or left (L-AAOCA), the latter including anomalies of just the left anterior descending or the left circumflex coronary artery.

AAOCA risk classification: This includes classifying the AAOCA risk as either low-risk anatomy (including the features of pre-pulmonic, retro-aortic, and high take-off) or high-risk anatomy (including the features of inter-arterial or sub-pulmonic courses).

The training of all models was conducted on the training dataset, where all model parameters and hyperparameters were established and used without any changes in the other datasets. For testing, we employed 3 different datasets (Figs. 1 and 7) to ensure robustness and generalizability. The internal testing dataset served as an internal

testing set, whereas external testing and clinical evaluation datasets provided real-world external test scenarios to assess model performance in different settings. For the coronary anomaly detection, the model was trained from scratch with a Kaiming weight initializer. For the other two classifications (origin and anatomical risk classification) models, which had smaller training datasets and included only the AAOCA data subset, we used the pre-trained network from the anomaly detection model.

**Evaluation of the models.** We did not apply any selection criteria related to sex or gender. Sex was gathered from registries based on patient self-reporting and CT DICOM image headers. No sex-specific restrictions were applied in the study design to ensure the generalizability of our findings. All analyses were disaggregated based on patient sex and reported in the supplemental datasets. Performance evaluations for all models were carried out using a variety of metrics in the internal and external testing datasets. We report different classification metrics, such as area under the curve (AUC), sensitivity, and specificity for each task. Due to the absence of labeled AAOCA cases in the external clinical evaluation dataset (Guangdong Provincial People's Hospital), we were able to evaluate the model's screening performance for anomaly detection, origin, and risk classification in real-world scenarios. We applied the same preprocessing steps and used the trained model on the entire dataset. We then selected cases flagged as AAOCA by the AI models. These cases were subsequently reviewed by two cardiologists experienced in CCTA. A final report was compiled based on these expert assessments to provide a detailed evaluation of the model's performance in detecting and classifying AAOCA in a real-world setting without prior labeling. Figure 1 presents a summary of model development and evaluation throughout the entire study. Supplementary Fig. 27 illustrates different strategies for model development and evaluation, which were implemented within our dataset using more labeled data in the model development (see the supplemental methods for more details). Supplementary Fig. 28 shows the different options for using the developed model in real clinical settings, from fully automated to semi-automated (physician in the loop) approaches.

**Ensembling, interpretability, feature visualization.** Given the numerous possibilities for network architectures and configurations that could be explored, we selected our model based on initial experiments conducted solely on the training and internal validation sets. To ensure the reproducibility of our results and avoid false discovery considering the limited computational power, we reported the model performances across five different trainings named 5-fold models (the 5 different models were trained using the same training–(internal)-validation split with different initialization in the weights). Then, we obtained an ensemble model whose predictions are the average of the single models' predictions. Moreover, to make the models more interpretable and to understand the features influencing their decisions, we implemented the Grad-CAM + +[29] (Gradient-weighted class activation mapping) algorithm in different blocks of the network. This allowed us to visually highlight which areas of the images were most significant in determining the outcomes. Finally, we used t-distributed Stochastic Neighbor Embedding (t-SNE) to visually represent the dataset based on the embedding features learned by the model and to see how these features discriminate between different data classes.

### Reporting summary
Further information on research design is available in the Nature Portfolio Reporting Summary linked to this article.

## Data availability
The dataset used for segmentation model development and clinical evaluation is publicly available[24] (Link: https://zenodo.org/records/

6802614). The fully anonymized datasets from Bern and Zurich allow restricted access only, in accordance with the requirements of the institutional review board (IRB) approvals and data sharing regulations. The raw datasets from Bern and Zurich University are protected and are not available due to data privacy laws. Access can be obtained upon IRB and Data Sharing Committee approvals from Bern, Zurich, and the requesting institution, within a time frame of one year. Details on how to request access are available from Dr. Christoph Gräni. The dataset from Guangdong Provincial People's Hospital (external clinical evaluation dataset), which can be used to test and evaluate different segmentation and classification models, is publicly available in ref. 23(Link: https://www.kaggle.com/datasets/xiaoweixumedicalai/imagecas). Corresponding code for the source data, to regenerate the figures and tables are also provided with this paper. Source data are provided in this paper.

## Code availability
All developed code and models are made publicly available on our AI-CVI laboratory's GitHub page (https://github.com/AI-in-Cardiovascular-Medicine/AAOCA) under the Attribution-NonCommercial 4.0 International (CC BY-NC 4.0) licenses[30]. In addition, we have also provided a publicly available web service, accessible via the following link (Link to the project: https://mb-neuro.medical-blocks.ch/public_access/projects and link to the WebApp: https://mb-neuro.medical-blocks.ch/public_access/projects/aaoca), which allows users to easily upload images in various desired formats for use to get the report and result based on models developed in the current study.

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

## Acknowledgements

This work was supported by the Swiss National Science Foundation (grant number 200871), Swiss Heart Foundation (grant number: FF23069), Novartis Foundation for medical-biological Research (grant number: 23B108), and the GAMBIT Foundation, all received by Christoph Gräni. We thank Laura Morf, Lukas Lüthi, Karini Ampalam, and Lea Rebecca Zurbriggen from the research study team for their excellent technical and administrative support. We also thank Martin Zbinden for his excellent technical engineering support.

## Author contributions

All authors made significant contributions to this research, fulfilling the criteria for authorship. M.R.B., A.W.S., R.K., M.S., S.W., L.R., A.A.G., G.C. M.S., R.R.B., and C.G. were responsible for raw material collection and data preparation from different databases and registries. W.V. and I.S. handled image preparation and preprocessing. I.S., G.B., and P.M.K. conducted all AI model development and evaluation. C.G. supervised the study. All authors reviewed, provided feedback on, and approved the final manuscript.

## Competing interests

Dr. Gräni received funding from the Swiss National Science Foundation (Grant number 200871), GAMBIT Foundation, Novartis Foundation for Medical-Biological Research, and Swiss Heart Foundation. The remaining authors declare that they have no competing interests in relation to the work described in this manuscript.
