## [Transparent Peer Review file · Nature Communications]

AI Based Anomalous Aortic Origin of Coronary Arteries Detection and Classification in Coronary CT Angiography

Corresponding Author: Professor Christoph Gräni

Version 0:

Reviewer comments:

Reviewer #1

(Remarks to the Author)

The authors present a manuscript investigating the development and validation of an AI-based tool for detecting and classifying anomalous aortic origin of the coronary artery (AAOCA) in 3D CCTA images. The study employs a multi-stage validation process—internal, retrospective and prospective, external, and real-world clinical testing—demonstrating the tool's robust performance and potential clinical impact. The topic is very important as there is a need for development of AI tools to identify high risk AAOCA. Even though AAOCA can be properly assessed by expert readers in large academic institutions, it presents significant challenges for less experienced readers. With the expansion of cardiac CT, the development of AI tools to aid less experienced readers becomes paramount. The study's methodology is thorough, ensuring the development of a reliable AI tool. It starts with internal validation, which provides a solid foundation for model development. Following this, the tool is tested on a prospective dataset, to assess performance on newly acquired data. This prospective testing ensures that the tool is not overfitted to historical data and remains effective in newly acquired patient data. The addition of external testing datasets confirms the model's robustness across different populations and imaging protocols. The use of an external, unlabeled clinical evaluation dataset highlights the tool's practical applicability in real-world settings.

A key strength is the use of data augmentation techniques to handle image variability across scanners and centers. By simulating motion artifacts and various acquisition methods, including older scanner generations, the model's robustness and generalizability are enhanced.

There are a few points that need clarification.

1. Abstract: Well-written and comprehensive.
2. Methods: Please clarify the specific AAOCA characteristics that were labeled as high risk.
3. Results:
 - Please include the number of cases with each high-risk feature.
 - Please include the different scanners used for image acquisition.
 - Where any cases excluded?
4. Discussion: Clear and comprehensive.
5. Conclusion: Supported by findings.

Minor comments:

- Line 110, there seems to be a typo on the number.
- Figure 5, low risk example does not show the origin of the coronaries.

Reviewer #2

(Remarks to the Author)

The manuscript by Shiri et al describes the development, training, testing and validation of a deep-learning AI model for identifying anomalous aortic origin of coronary arteries (AAOCA) on coronary CT angiography. The goal of the model is to

improve identification of anomalous coronary arteries on CCTA, for both clinical screening and research purposes. They used both internal and external 3D-CCTA images to serve as the training and testing datasets, and provide a comprehensive description of their methodology, and the discriminatory performance and accuracy of the ultimate trained model. The accuracy was excellent with sensitivity and specificity in general >0.95 for identification of AAOCA, origin classification, and assessment for high-risk features. Notably, given the low prevalence of disease in the external "clinical evaluation" dataset, my understanding from the reported results is that a high proportion (20/24) of the identified AAOCA were false positives (low positive predictive value).

Overall, the manuscript is well written, with strong AI-based methods, comprehensive training and testing in datasets from multiple different centers, and addresses a relevant topic: the use of advanced AI methods in medical imaging and specifically CCTA to aid in the identification of disease. I can certainly see the strong value of using this approach in CCTA research registries. However, given the low PPV in the clinical dataset, it is somewhat questionable whether AI could replace an expert clinical reader, as any identified cases would need to be carefully reviewed for accurate identification by an expert. In low-volume centers with less experienced readers, it could lead to false positive reports and potential patient anxiety/harm. However, for expert readers in high-volume centers where time constraints are always present, it could be an effective safety check to ensure no cases of anomalous coronaries are missed inadvertently. As the authors point out, with larger training sets, the false positive rate went down, so the AI model might need to be further trained to enter the clinical arena. With that caveat, this is a strong research paper that would add value to the CCTA literature and serve as a model for other AI-approaches.

I have a few suggestions:

1. I would modify the discussion section to reflect the limitations of use in clinical imaging, due to the low PPV in the external clinical evaluation dataset, and implications for use in smaller centers especially, with less expertise to discriminate false positives. For instance, would modify the following statement:

a. "Additionally, the difficulty in correctly classifying anomalies as potentially high risks versus low risks needs expertise, which may be lacking in smaller centers providing CCTA programs. Consequently, this AI tool could help to real time alert physicians about AAOCA and its potential risk."

2. The methods section is long and could use some reorganization. Might consider moving some or all of the discussion of the three strategies (pages 8-9 line 226-253) to the supplementary material. It distracts from the readability, is confusing, and I don't think adds a lot to the message of the paper since strategy 1 is primarily reported. There could still be brief mention in the main text that using a larger training set (all datasets with labels) decreased false positives, and refer readers to supplementary materials.

3. It would be helpful to know more details about the CCTA scanners and protocols used in the training and testing datasets (maybe in supplemental material), which would inform generalizability to other centers. What were the year ranges? What were the scanners/scanner technology used in the datasets? Was it systolic or diastolic phase imaging? Was it only coronary CT with administration of nitroglycerin, or would this work on non-coronary protocol cardiac CT?

Minor Issues

1. Review introduction and discussion for minor clarity issues and typographic errors

Reviewer #3

(Remarks to the Author)

Version 1:

Reviewer comments:

Reviewer #1

(Remarks to the Author)

The authors have thoroughly and appropriately addressed all comments.

(Remarks on code availability)

Reviewer #2

(Remarks to the Author)

The revisions addressed all of my concerns.

(Remarks on code availability)

Reviewer #3

(Remarks to the Author)

(Remarks on code availability)

Action Summary for Comments from the Reviewer 1

The authors present a manuscript investigating the development and validation of an AI-based tool for detecting and classifying anomalous aortic origin of the coronary artery (AAOCA) in 3D CCTA images. The study employs a multi-stage validation process—internal, retrospective and prospective, external, and real-world clinical testing—demonstrating the tool's robust performance and potential clinical impact. The topic is very important as there is a need for development of AI tools to identify high risk AAOCA. Even though AAOCA can be properly assessed by expert readers in large academic institutions, it presents significant challenges for less experienced readers. With the expansion of cardiac CT, the development of AI tools to aid less experienced readers becomes paramount. The study's methodology is thorough, ensuring the development of a reliable AI tool. It starts with internal validation, which provides a solid foundation for model development. Following this, the tool is tested on a prospective dataset, to assess performance on newly acquired data. This prospective testing ensures that the tool is not overfitted to historical data and remains effective in newly acquired patient data. The addition of external testing datasets confirms the model's robustness across different populations and imaging protocols. The use of an external, unlabeled clinical evaluation dataset highlights the tool's practical applicability in real-world settings.

A key strength is the use of data augmentation techniques to handle image variability across scanners and centers. By simulating motion artifacts and various acquisition methods, including older scanner generations, the model's robustness and generalizability are enhanced.

Authors: The authors would like to thank the reviewer for her/his positive comments. We would like to note that we have made all the code developed in this study publicly available on GitHub [link: <https://github.com/AI-in-Cardiovascular-Medicine/AAOCA>]. Additionally, the models are accessible via the following link: [link: <https://mb-neuro.medical-blocks.ch/shared/file/bcf0feb0-b182-11ef-8549-91b9ef42f351>]. Moreover, the code is provided in a zip file, which includes a README with installation and usage instructions. To simplify implementation, we have provided a Docker file [link: https://github.com/AI-in-Cardiovascular-Medicine/AAOCA/tree/main/docker_project] and a Docker image [link: <https://mb-neuro.medical-blocks.ch/shared/file/f797b5f0-bc5d-11ef-8de9-d7709cd3a54c>], both of which can be easily downloaded and installed. If the direct access to the links does not work, please copy and paste them into your browser.

To further enhance usability, we have developed a publicly accessible web service that allows users to upload images in various formats and receive reports and results based on the models developed in this study. The web service can be accessed here: [link: https://mb-neuro.medical-blocks.ch/public_access/projects], with direct access to the WebApp via this link: [link: https://mb-neuro.medical-blocks.ch/public_access/projects/aaoca]. We have also provided examples [link: <https://mb-neuro.medical-blocks.ch/shared/folder/67a394b0-ae38-11ef-89f5-5137b14d4312>] with expected outcomes to help reviewers test the code, models, and WebApp. Additionally, the external clinical evaluation dataset from Guangdong Provincial People's Hospital, consisting of 1000 patient datasets (1000 3D CT images), is publicly available on Kaggle (link: <https://www.kaggle.com/datasets/xiaoweixumedicallai/imagecas>) and can be

used to evaluate different segmentation and classification models in different format of codes, Docker and also the WebApp.

To access the WebApp, users must request a code via the correct email address to ensure server security and prevent potential attacks by bots. To simplify this process, we are providing the following email and code for reviewers and editors to bypass registration restrictions:

Email: aaoca@medical-blocks.ch

Code: LADpRJAp9XHTxrHX0Bd2X

Alternatively, you may register independently to obtain your own codes if preferred. Another point relates to inference time, as detailed in the *Inference Time* section of our GitHub page [Link: <https://github.com/AI-in-Cardiovascular-Medicine/AAOCA>]. As mentioned in the manuscript, for each task, we utilize an ensemble of different models (5 models per individual task, and we have four tasks). The average running time for processing—from raw image input to final output—on a standard desktop PC (Core i9 processor, 3200 MHz with one NVIDIA 4090 GPU) is approximately 27 seconds using the recommended GPU setup. This configuration is consistent with the reported results in the manuscript. However, the inference time can be reduced to just a few seconds by using fewer models per task. On WebApp, inference time varies due to hardware specifications, such as the CPU (2000 MHz) version and the size of the uploaded images. It typically ranges from a few seconds to up to 2-3 minutes as we run all 20 models in the background to replicate the exact results reported in the manuscript. We highly recommend using the main models and code provided on GitHub using a local PC with GPU for a comprehensive evaluation of the models developed in this study and their inference times. While the results of WebApp and codes will be identical, the computational resources, including the GPU, CPU, and Memory on our server, are limited. Therefore, we suggest running the models locally, especially for scenarios not tested in this study. We are currently in the process of upgrading our server hardware, and the next version of the application is expected to achieve significantly reduced inference times, running in just a few seconds. We remain open to any further modifications suggested by the reviewer.

There are a few points that need clarification.

Comment #1: 1. Abstract: Well-written and comprehensive.

Response #1: We thank the reviewer for his/her positive feedback. Following the journal guidelines, we have condensed the abstract to a maximum of 150 words from the original 425, aiming to maintain the essence of the previous version.

Comment #2: 2. Methods: Please clarify the specific AAOCA characteristics that were labeled as high risk.

Response #2: We have now clarified the specific AAOCA characteristics labeled as high risk within the dataset.

“AAOCA risk classification: This includes classifying the AAOCA risk as either low-risk anatomy (including the features of pre-pulmonic, retro-aortic, and high take-off) or high-risk anatomy (including the features of inter-arterial or sub-pulmonic courses).”

3. Results:

Comment #3: -Please include the number of cases with each high-risk feature.

Response #3: We have now provided the number of cases with high-risk features across the different datasets analyzed.

“Out of 225 CCTA high-risk anatomy AAOCA, 208 patients (773 CCTA images) had an inter-arterial course, and 17 patients (47 CCTA images) had a sub-pulmonic course.”

Comment #4: -Please include the different scanners used for image acquisition.

Response #4: We have now provided additional information on data acquisition, including the scanners and acquisition parameters used for each dataset, in the supplemental material. Please see the **Supplemental Tables 2 and 3**.

Comment #5: -Where any cases excluded?

Response #5: We thank the reviewer for this comment. We excluded the following cases and clarified this in the manuscript:

In Methods we specify:

“From all datasets, we excluded patients without contrast-enhanced cardiac CT images, those with uninterpretable images due to severe artifacts, and images with cropped regions of the aorta. From the datasets used for training and testing, we excluded from the analysis those without a report for the origin and anatomical risk classification.”

In Results we specify:

“Overall, we excluded from the analysis 14 patients without contrast-enhanced cardiac CT images, 10 with uninterpretable images due to severe artifacts, 6 patients with images with cropped regions of the aorta, and 12 without a report for the origin and anatomical risk classification.”

Comment #6: 4. Discussion: Clear and comprehensive.

5. Conclusion: Supported by findings.

Response #6: We thank the reviewer for this positive feedback. In response to the editor’s and reviewers’ comments, we have updated various sections of the discussion, highlighted in red. We are also open to any further suggestions from the reviewer to enhance the revised abstract and ensure it meets the criteria.

Minor comments:

Comment #7: -Line 110, there seems to be a typo on the number.

Response #7: We thank the reviewer for this point. We have now reviewed the manuscript and addressed grammatical issues, including typographical errors.

Comment #8: -Figure 5, low risk example does not show the origin of the coronaries.

Response #8: We thank the reviewer for this point. In this figure, for the low-risk example with a pre-pulmonic course, our aim is to illustrate the course of the coronary artery. Due to the anatomy of the coronary artery, it is not possible to show both the origin and the full course in a single slice even with reorientation (we can observe that the continuous course cannot be displayed in a single slice). Therefore, in each example, we focused either on the origin for origin classification or the course of the coronary artery for both low- and high-risk cases. However, to address this comment, we provided a gif video of the image and GradCam overly in our Git Hub (explainability section) for different cases which shows the origin and the course in different cases. We are open to providing any images or information if the reviewer requests it.

Action Summary for Comments from the Reviewer 2

The manuscript by Shiri et al describes the development, training, testing and validation of a deep-learning AI model for identifying anomalous aortic origin of coronary arteries (AAOCA) on coronary CT angiography. The goal of the model is to improve identification of anomalous coronary arteries on CCTA, for both clinical screening and research purposes. They used both internal and external 3D-CCTA images to serve as the training and testing datasets, and provide a comprehensive description of their methodology, and the discriminatory performance and accuracy of the ultimate trained model. The accuracy was excellent with sensitivity and specificity in general >0.95 for identification of AAOCA, origin classification, and assessment for high-risk features. Notably, given the low prevalence of disease in the external “clinical evaluation” dataset, my understanding from the reported results is that a high proportion (20/24) of the identified AAOCA were false positives (low positive predictive value).

Overall, the manuscript is well written, with strong AI-based methods, comprehensive training and testing in datasets from multiple different centers, and addresses a relevant topic: the use of advanced AI methods in medical imaging and specifically CCTA to aid in the identification of disease. I can certainly see the strong value of using this approach in CCTA research registries. However, given the low PPV in the clinical dataset, it is somewhat questionable whether AI could replace an expert clinical reader, as any identified cases would need to be carefully reviewed for accurate identification by an expert. In low-volume centers with less experienced readers, it could lead to false positive reports and potential patient anxiety/harm. However, for expert readers in high-volume centers where time constraints are always present, it could be an effective safety check to ensure no cases of anomalous coronaries are missed inadvertently. As the authors point out, with larger training sets, the false positive rate went down, so the AI model might need to be further trained to enter the clinical arena. With that caveat, this is a strong research paper that would add value to the CCTA literature and serve as a model for other AI-approaches.

Authors: The authors would like to take this opportunity to thank the reviewer for her/his constructive criticisms, suggestions, and positive feedback. These comments have contributed to remarkably improve the overall quality of the manuscript. We addressed all of the above-mentioned comments and suggestions in the revised manuscript.

As we mentioned in our responses to the Editors, we have made all the code developed in this study publicly available on GitHub [link: <https://github.com/AI-in-Cardiovascular-Medicine/AAOCA>]. Additionally, the models are accessible via the following link: [link: <https://mb-neuro.medical-blocks.ch/shared/file/bcf0feb0-b182-11ef-8549-91b9ef42f351>]. Moreover, the code is provided in a zip file, which includes a README with installation and usage instructions. To simplify implementation, we have provided a Docker file [link: https://github.com/AI-in-Cardiovascular-Medicine/AAOCA/tree/main/docker_project] and a Docker image [link: <https://mb-neuro.medical-blocks.ch/shared/file/f797b5f0-bc5d-11ef-8de9-d7709cd3a54c>], both of which can be easily downloaded and installed. As mentioned by the reviewer, with these open access codes and models, the AI model might be further improved to enter the clinical arena.

To further enhance usability, we have developed a publicly accessible web service that allows users to upload images in various formats and receive reports and results based on the models developed in this study. The web service can be accessed here: [link: https://mb-neuro.medical-blocks.ch/public_access/projects], with direct access to the WebApp via this link: [link: https://mb-neuro.medical-blocks.ch/public_access/projects/aaoca]. We have also provided examples [link: <https://mb-neuro.medical-blocks.ch/shared/folder/67a394b0-ae38-11ef-89f5-5137b14d4312>] with expected outcomes to help reviewers test the code, models, and WebApp. Additionally, the external clinical evaluation dataset from Guangdong Provincial People’s Hospital, consisting of 1000 patient datasets (1000 3D CT images), is publicly available on Kaggle (link: <https://www.kaggle.com/datasets/xiaoweixumedicallai/imagecas>) and can be used to evaluate different segmentation and classification models in different format of codes, Docker and also the WebApp.

To access the WebApp, users must request a code via the correct email address to ensure server security and prevent potential attacks by bots. To simplify this process, we are providing the following email and code for reviewers and editors to bypass registration restrictions:

Email: `aaoca@medical-blocks.ch`

Code: `LADpRJAp9XHTxrHX0Bd2X`

Alternatively, you may register independently to obtain your own codes if preferred. If direct access to the links does not work, please copy and paste them into your browser. Another point relates to inference time, as detailed in the *Inference Time* section of our GitHub page [Link: <https://github.com/AI-in-Cardiovascular-Medicine/AAOCA>]. As mentioned in the manuscript, for each task, we utilize an ensemble of different models (5 models per individual task, and we have four tasks). The average running time for processing—from raw image input to final output—on a standard desktop PC (Core i9 processor, 3200 MHz with one NVIDIA 4090 GPU) is approximately 27 seconds using the recommended GPU setup. This configuration is consistent with the reported results in the manuscript. However, the inference time can be reduced to just a few seconds by using fewer models per task. On WebApp, inference time varies due to hardware specifications, such as the CPU (2000 MHz) version and the size of the uploaded images. It typically ranges from a few seconds to up to 2-3 minutes as we run all 20 models in the background to replicate the exact results reported in the manuscript. We highly recommend using the main models and code provided on GitHub using a local PC with GPU for a comprehensive evaluation of the models developed in this study and their inference times. While the results of WebApp and codes will be identical, the computational resources, including the GPU, CPU, and Memory on our server, are limited. Therefore, we suggest running the models locally, especially for scenarios not tested in this study. We are currently in the process of upgrading our server hardware, and the next version of the application is expected to achieve significantly reduced inference times, running in just a few seconds. We remain open to any further modifications suggested by the reviewer. We remain open to any further modifications suggested by the reviewer.

I have a few suggestions:

Comment #1: 1. I would modify the discussion section to reflect the limitations of use in clinical imaging, due to the low PPV in the external clinical evaluation dataset, and implications for use in smaller centers especially, with less expertise to discriminate false positives. For instance, would modify the following statement:

a. “Additionally, the difficulty in correctly classifying anomalies as potentially high risks versus low risks needs expertise, which may be lacking in smaller centers providing CCTA programs. Consequently, this AI tool could help to real time alert physicians about AAOCA and its potential risk.”

Response #1: We thank the reviewer for these insightful remarks, and we agree with all points raised. However, we would like to clarify a few aspects in this response. First, our study aimed to detect AAOCA cases (positive cases) as accurately as possible within the performance limits of the AI model. Given the rarity of AAOCA, minimizing missed cases is crucial, emphasizing the importance of accurate true positive detection. In our clinical dataset of 998 cases, Strategy 1 produced only 20 false positives (2%), which decreased to 9 cases (0.9%) in Strategy 3, without any change in true positive detections (4 in all). This consistency reflects the robustness of the model for AAOCA detection, which is clinically significant and decreases the false positive with more training datasets, as pointed out by the reviewer.

Additionally, with an unbalanced clinical dataset—where only 0.4% of cases (4 out of 998) are positive—the PPV may not fully represent model performance. This is one reason we did not report this metric in the unlabeled dataset; instead, we focused on the number of detected cases, where positive AAOCA detections remained stable across strategies, while false positives dropped from 2% to 0.9%. Notably, the PPV in the internal and external test sets, which have more balanced datasets, exceeds 97%. Clinically, true positive detection of AAOCA is critical, but a model with reasonable false positives is also important. In our case, even in an external clinical dataset, the false positive rate was manageable, reducing the cases requiring physician review from 998 to only 24 in Strategy 1 (4 true positives and 20 false positives) and 13 in Strategy 3 (4 true positives and 9 false positives). Moreover, we illustrated in Figure 7 that some false positive cases were highly challenging, even for clinicians, with explanations provided for each case in the figure captions. In addition, we provided the results with different cut-offs now (0.1 to 0.9), which show the compromise between false positive, false negative, true negative, and true positive. Based on these threshold results, users can select an optimal threshold specified to their use case—for instance, some may prioritize high sensitivity, while others may prefer high specificity. Furthermore, we have implemented this threshold adjustment functionality in the WebApp, allowing users to modify it as needed when applying the model to new cases.

Finally, we have emphasized that our model is not intended to replace clinical expertise but to augment it. Moreover, we mentioned this in the WebApp and GitHub as “*This software is intended for research purposes only and has not been approved for clinical use.*”. We fully agree with the reviewer that expanding datasets can further reduce the false positive rate, as demonstrated across strategies. We believe that with increased data, along with a high true

positive rate, the model's false positive rate can be further minimized. Moreover, we have made all our developed codes [link: <https://github.com/AI-in-Cardiovascular-Medicine/AAOCA>] and models [link: <https://mb-neuro.medical-blocks.ch/shared/file/bcf0feb0-b182-11ef-8549-91b9ef42f351>] with instructions to run (train and test) it publicly available, enabling researchers to use them as a backbone for their studies to train further and enhance model performance, including reducing false positives and facilitating their entry into the clinical arena.

Based on this comment and the previous comment, we modified the corresponding text and added the following paragraphs:

“While this AI tool has the potential to assist by alerting physicians in real-time to the presence of AAOCA and its potential risks, in smaller centers with less experienced readers, reliance on AI without expert oversight might lead to false positives, causing unnecessary follow-ups, increased patient anxiety, and potential harm. In high-volume centers with experienced readers, the tool could serve as an effective safety check to ensure no cases of anomalous coronaries are missed. Furthermore, the current model is designed to augment—not replace—clinical expertise, particularly in settings where additional support can aid in efficiently identifying AAOCA cases.”

Moreover, we added the following for further clarification:

" Given the rarity of AAOCA, minimizing missed cases is crucial, emphasizing the importance of accurate true positive detection. While false positives were observed in the external clinical test set, they were significantly reduced from 2% to 0.9% through more training datasets without compromising true positive detections (from strategies 1 to 3). As our study demonstrates, further training on larger datasets could reduce false positives even further. Future research should focus on enhancing these models to minimize false positives and facilitate their integration into clinical practice. "

Comment #2: 2. The methods section is long and could use some reorganization. Might consider moving some or all of the discussion of the three strategies (pages 8-9 line 226-253) to the supplementary material. It distracts from the readability, is confusing, and I don't think adds a lot to the message of the paper since strategy 1 is primarily reported. There could still be brief mention in the main text that using a larger training set (all datasets with labels) decreased false positives, and refer readers to supplementary materials.

Response #2: We thank the reviewer for this suggestion. We have now shortened the Methods section and moved parts of the Methods and Discussion to the supplemental material to improve readability. Specifically, we moved the mentioned section to the supplemental material and briefly noted in the main text that using a large training set reduces false positives, directing readers to the supplemental materials for more details.

We added the following for more clarification:

" Given the rarity of AAOCA, minimizing missed cases is crucial, emphasizing the importance of accurate true positive detection. While false positives were observed in the external clinical

test set, they were significantly reduced from 2% to 0.9% through more training datasets without compromising true positive detections (from strategies 1 to 3). As our study demonstrates, further training on larger datasets could reduce false positives even further. Future research should focus on enhancing these models to minimize false positives and facilitate their integration into clinical practice. "

Comment #3: 3. It would be helpful to know more details about the CCTA scanners and protocols used in the training and testing datasets (maybe in supplemental material), which would inform generalizability to other centers. What were the year ranges? What were the scanners/scanner technology used in the datasets? Was it systolic or diastolic phase imaging? Was it only coronary CT with administration of nitroglycerin, or would this work on non-coronary protocol cardiac CT?

Response #3: We thank the reviewer for this comment. We have now added more details on the CCTA scanners, acquisition, and reconstruction parameters across different datasets to support generalizability to other centers. Additionally, we have included information on the range of acquisition years, scanner names, and technologies used. Please see the supplemental tables 2 and 3.

Our study utilized different phases of cardiac images, including both systolic and diastolic phases, to ensure robustness and applicability across various CT images. For patients with 4D CT, we used multiple phases, which function as natural data augmentation to increase both the dataset size and generalizability (considering that data splitting was performed patients-wise, not image-wise, which means all images from one patient are always in one cohort to avoid any data leakage). The model was developed and evaluated using contrast-enhanced cardiac CT images; we did not evaluate it on non-coronary protocol CT images. However, we hypothesize that if images are contrast-enhanced with sufficient resolution to display the coronary arteries, the model could potentially perform well. Since this hypothesis was not tested, we have noted it in the discussion section as an area for future evaluation.

We added following in the method section:

"In cases where different phases of cardiac images were available, we utilized all available reconstructed phases as a form of natural data augmentation to increase both the dataset size and the model's generalizability."

We added the following in the discussion section:

"The model development and evaluation were performed using CCTA images. However, with further testing, this model could potentially be adapted for non-coronary contrast-enhanced CT images in future studies, broadening its applicability to other imaging acquisition and protocols toward enhancing the AAOCA detection. Developing a deep learning model that detects anomalies in non-contrast images, such as standard chest CTs, would expand its utility beyond cardiac imaging to include applications like any chest CT scans. However, detecting coronary arteries in non-contrast images presents significant challenges due to low image

contrast and possibly low resolution, complicating the analysis even for highly experienced cardiologists and radiologists..”

Minor Issues

Comment #4: 1. Review introduction and discussion for minor clarity issues and typographic errors

Response #4: We thank the reviewer for this point. We have now reviewed the manuscript and addressed grammatical issues, including typographical errors.

Action Summary for Comments from the Reviewer 3

Authors: The authors would like to take this opportunity to thank the reviewer for her/his efforts and constructive criticisms and suggestions. Those comments have contributed to remarkably improving the overall quality of the manuscript. We believe all of them were satisfied in the enclosed revised version.

We have made all the code developed in this study publicly available on GitHub [link: <https://github.com/AI-in-Cardiovascular-Medicine/AAOCA>]. Additionally, the models are accessible via the following link: [link: <https://mb-neuro.medical-blocks.ch/shared/file/bcf0feb0-b182-11ef-8549-91b9ef42f351>]. Moreover, the code is provided in a zip file, which includes a README with installation and usage instructions. To simplify implementation, we have provided a Docker file [link: https://github.com/AI-in-Cardiovascular-Medicine/AAOCA/tree/main/docker_project] and a Docker image [link: <https://mb-neuro.medical-blocks.ch/shared/file/f797b5f0-bc5d-11ef-8de9-d7709cd3a54c>], both of which can be easily downloaded and installed. If the direct access to the links does not work, please copy and paste them into your browser.

To further enhance usability, we have developed a publicly accessible web service that allows users to upload images in various formats and receive reports and results based on the models developed in this study. The web service can be accessed here: [link: https://mb-neuro.medical-blocks.ch/public_access/projects], with direct access to the WebApp via this link: [link: https://mb-neuro.medical-blocks.ch/public_access/projects/aaoca]. We have also provided examples [link: <https://mb-neuro.medical-blocks.ch/shared/folder/67a394b0-ae38-11ef-89f5-5137b14d4312>] with expected outcomes to help reviewers test the code, models, and WebApp. Additionally, the external clinical evaluation dataset from Guangdong Provincial People's Hospital, consisting of 1000 patient datasets (1000 3D CT images), is publicly available on Kaggle (link: <https://www.kaggle.com/datasets/xiaoweixumedicallai/imagecas>) and can be used to evaluate different segmentation and classification models in different format of codes, Docker and also the WebApp.

To access the WebApp, users must request a code via the correct email address to ensure server security and prevent potential attacks by bots. To simplify this process, we are providing the following email and code for reviewers and editors to bypass registration restrictions:

Email: aaoca@medical-blocks.ch

Code: LADpRJAp9XHTxrHX0Bd2X

Alternatively, you may register independently to obtain your own codes if preferred. Another point relates to inference time, as detailed in the *Inference Time* section of our GitHub page [Link: <https://github.com/AI-in-Cardiovascular-Medicine/AAOCA>]. As mentioned in the

manuscript, for each task, we utilize an ensemble of different models (5 models per individual task, and we have four tasks). The average running time for processing—from raw image input to final output—on a standard desktop PC (Core i9 processor, 3200 MHz with one NVIDIA 4090 GPU) is approximately 27 seconds using the recommended GPU setup. This configuration is consistent with the reported results in the manuscript. However, the inference time can be reduced to just a few seconds by using fewer models per task. On WebApp, inference time varies due to hardware specifications, such as the CPU (2000 MHz) version and the size of the uploaded images. It typically ranges from a few seconds to up to 2-3 minutes as we run all 20 models in the background to replicate the exact results reported in the manuscript. We highly recommend using the main models and code provided on GitHub using a local PC with GPU for a comprehensive evaluation of the models developed in this study and their inference times. While the results of WebApp and codes will be identical, the computational resources, including the GPU, CPU, and Memory on our server, are limited. Therefore, we suggest running the models locally, especially for scenarios not tested in this study. We are currently in the process of upgrading our server hardware, and the next version of the application is expected to achieve significantly reduced inference times, running in just a few seconds. We remain open to any further modifications suggested by the reviewer.

Action Summary for Comments from the Reviewers 1, 2, 3

Reviewer #1 (Remarks to the Author): The authors have thoroughly and appropriately addressed all comments.

Reviewer #2 (Remarks to the Author): The revisions addressed all of my concerns.

Reviewer #3 (Remarks to the Author): I co-reviewed this manuscript with one of the reviewers who provided the listed reports. This is part of the Nature Communications initiative to facilitate training in peer review and to provide appropriate recognition for Early Career Researchers who co-review manuscripts.

Authors: The authors sincerely thank the reviewers for their valuable contributions and insightful feedback, which have played a significant role in improving the manuscript throughout the revision process.